# Exploring the Origins of Association of Poly(acrylic acid) Polyelectrolyte with Lysozyme in Aqueous Environment through Molecular Simulations and Experiments

**DOI:** 10.3390/polym16182565

**Published:** 2024-09-11

**Authors:** Maria Arnittali, Sokratis N. Tegopoulos, Apostolos Kyritsis, Vagelis Harmandaris, Aristeidis Papagiannopoulos, Anastassia N. Rissanou

**Affiliations:** 1Institute of Applied and Computational Mathematics, Foundation for Research and Technology Hellas, IACM/FORTH, GR-71110 Heraklion, Greece; maria.arnittali@gmail.com (M.A.); harman@uoc.gr (V.H.); 2Department of Mathematics and Applied Mathematics, University of Crete, GR-71409 Heraklion, Greece; 3Computation-Based Science and Technology Research Center, The Cyprus Institute, Nicosia 2121, Cyprus; 4School of Applied Mathematical and Physical Sciences, National Technical University of Athens, GR-15772 Athens, Greece; stegopoulos@mail.ntua.gr (S.N.T.); akyrits@central.ntua.gr (A.K.); 5Theoretical & Physical Chemistry Institute, National Hellenic Research Foundation, 48 Vassileos Constantinou Avenue, GR-11635 Athens, Greece

**Keywords:** polyelectrolytes, proteins, molecular dynamics simulations, secondary structure analysis, complexation, thermal treatment, Fourier transform infrared spectroscopy, circular dichroism

## Abstract

This study provides a detailed picture of how a protein (lysozyme) complexes with a poly(acrylic acid) polyelectrolyte (PAA) in water at the atomic level using a combination of all-atom molecular dynamics simulations and experiments. The effect of PAA and temperature on the protein’s structure is explored. The simulations reveal that a lysozyme’s structure is relatively stable except from local conformational changes induced by the presence of PAA and temperature increase. The effect of a specific thermal treatment on the complexation process is investigated, revealing both structural and energetic changes. Certain types of secondary structures (i.e., α-helix) are found to undergo a partially irreversible shift upon thermal treatment, which aligns qualitatively with experimental observations. This uncovers the origins of thermally induced aggregation of lysozyme with PAA and points to new PAA/lysozyme bonds that are formed and potentially enhance the stability in the complexes. As the temperature changes, distinct amino acids are found to exhibit the closest proximity to PAA, resulting into different PAA/lysozyme interactions; consequently, a different complexation pathway is followed. Energy calculations reveal the dominant role of electrostatic interactions. This detailed information can be useful for designing new biopolymer/protein materials and understanding protein function under immobilization of polyelectrolytes and upon mild denaturation processes.

## 1. Introduction

The application of green and biocompatible protocols for the creation of biomaterials appears as a crucial priority in recent literature [1]. Preparation of biomaterials using physical associations between biocompatible macromolecules opens a great variety of possibilities for the development of nanostructures for medical [2], food [3], and agricultural [4] industries. Electrostatically driven protein/polyelectrolyte complexation was extensively studied in the last three decades [5,6] and led to promising biomaterials, including nanoparticles for drug delivery [7,8], bioadhesive scaffolds for articular cartilage regeneration [9], and microcapsules for the entrapment and viability improvement of probiotic bacteria [10]. Proteins along with their specific functions offer multifunctionality as they have hydropathy and pH-tunable charge surface distributions [11]. Therefore, they can interact with electrically charged macromolecules and can encapsulate hydrophobic substances. Apart from the exploitation of self-association in such systems, the use of mild thermal treatments supports the development of new materials because biopolymers and especially proteins alter their conformations under temperature changes [12]. A well-documented case is the one of bovine serum albumin (BSA), which opens its conformation at elevated temperature to expose hydrophobic regions, which drive inter-protein associations by intermolecular β-sheets upon cooling back to room temperature [13]. This feature was used to stabilize chondroitin sulfate/BSA electrostatic complexes against disintegration at neutral pH [14]. Similar effects were utilized for β-lactoglobulin [15].

Lysozyme is often used as a model protein for studies of protein/polyelectrolyte complexes [16]. These complexes can adopt various structures (gel, microgel, micelle, colloid, precipitate, and coacervate) by adjusting the lysozyme/polyelectrolyte ratio as well as the nature and length of the polyelectrolyte [17,18,19,20]. Additionally, it was found that the impact on the structure and activity of lysozyme was determined by its interactions with polysaccharides. After thermal treatment, κ-carrageenan and konjac glucomannan were found to improve the stability of lysozyme-based complex systems [21]. FTIR spectroscopy revealed that κ-carrageenan induces the rise of β-structure in lysozyme [22].

The role and nature of molecular interactions in protein/polyelectrolyte complexes [23] is fundamental for designing and optimizing these materials. The effects of complexation and the temperature response on the protein conformation are open questions for understanding the structural arrangement of the building blocks in the complexes, their mode of interaction, the binding with other molecules and their response or tolerance to external stimuli. For this purpose, atomic information is crucial in order to shed light on the dominant interactions and the responsive protein sites. Simulations provide considerable insight into the structure and the behavior of complexes. The recognition of the polyelectrolyte–protein complex coacervate binding mechanism in terms of both conformational preferences and energetics is significant for fine tuning of suitable parameters towards the formation of complexes with specific properties for different applications.

Various computational approaches and models have been employed to address these important questions [24,25,26,27,28,29,30]. A very recent work focused on the effect of hydrophobicity on the sequence-dependent peptide conformation in polyelectrolyte complexes using molecular dynamics (MD) simulations [31]. The study revealed that the peptide conformation and degree of hydrogen bonding are influenced by the specific sequence of the peptides, showing sensible trends consistent with experimental results. The importance of sequence-dependent conformational properties and hydrogen bonding of proteins is highlighted in the work of Arnittali et al. as well [32]. In another study by Xu et al. [33], the interaction of proteins with polyelectrolytes was investigated using a combination of experimental work, simulations, and mean-field theories, where the important role of counterions was demonstrated [34]. Moreover the replica exchange technique [25] was utilized to explore the way that charged polymers affect the thermal stability of beta-hairpin peptides. Sofronova et al. [30] employed atomistic MD simulations to explore the complexation of the cationic protein lysozyme with poly(styrene sulfonate) and polyphosphate of different degree of polymerization and highly charged polyanions. It was observed that the shorter charged chains bound almost completely with the protein, while longer chains exhibit unbound terminal segments that adopt charged loop and tail conformations. Furthermore, Monte Carlo simulations using a simple model were used to examine the complexation between a single protein with a single polyelectrolyte [28,29]. Electrostatic interactions were found to be the major factor in forming these complexes, especially when the protein is highly charged at low ionic strength conditions. However, the study also showed that electrostatic attraction is not the only force driving these complexes to form.

These studies highlight the importance and necessity of detailed analysis at the microscopic level to understand the origin of key interactions between proteins and polyelectrolytes and to determine sequence–property relationships. Fully atomistic simulations provide the most comprehensive approach to achieve this, enabling a thorough exploration of how individual atoms contribute to a molecule’s specific behavior.

The current work is based on MD simulations of an aqueous solution of the biocompatible macromolecules poly(acrylic acid) (PAA) polyelectrolyte and lysozyme [35,36]. Studying their association could be useful for designing bioadhesives, tissue engineering scaffolds, or other biomedical applications that rely on interactions between polymers and proteins [37]. Furthermore, it is a simple model system for the understanding of fundamental interactions between charged polymers and proteins. This study delves into the atomic-level impact of PAA on lysozyme’s conformational properties and investigates the effects of temperature on the protein’s structure and potentially on its enzymatic activity. We comprehensively examine how a quenching process influences the conformations of both PAA chains and the protein within the complexes, which is connected to a biocompatible treatment that induces interprotein associations and enhances stability in protein-based systems. All-atom MD simulations with explicit solvent model and explicit ion representation are utilized. By analyzing the protein’s secondary structure and the relative positioning of the two molecules within the complex, we elucidate their structural arrangement (i.e., the recognition of polyelectrolyte–protein complex coacervate binding sites). Energetic interactions, including electrostatic, Van der Waals, and hydrogen bonding components, reveal the primary driving force for complex formation and quantify the binding affinity between PAA and lysozyme. All these measures highlight short-range conformational changes of the protein molecules and validate the stability of the complex and the temperature-dependent reversibility of the complexation process. This detailed microscopic characterization, which to the best of our knowledge is performed for the first time, provides a comprehensive understanding of the factors governing the system’s behavior under specific conditions. This approach is essential for designing and optimizing biomaterials with precisely controlled properties.

## 2. Systems and Methods

### 2.1. Simulation Details

One lysozyme molecule in water comprised the reference system while the mixture included one protein molecule and 5 PAA chains (molecules), each one consisted of 40 monomer units. Details for the simulated systems are presented in Table 1. Simulations of 200 ns were performed at a series of temperatures: T = [298, 308, 328, 348, 358, and 368] K. The mass ratio in the mixture r_m_ = c_PAA_/c_Lysozyme_ was 1. The pKa of PAA is about 4.5 [38]. The distance between successive carboxylic units along the PAA chain is 0.27 nm and the Bjerrum length in water is 0.7 nm [39]. On this basis, half of the carboxylic groups of the PAA chain were deprotonated (i.e., 50% dissociation was assumed to account for counterion condensation). The positive charge of the lysozyme was counterbalanced by deprotonated carboxylic groups of PAA while sodium counterions were used to ensure total electrical neutrality rendering an ionic strength I = 0.15 M for the solution. In the initial configuration, all molecules were well-dispersed in the aqueous environment.

All-atom molecular dynamics simulations were performed using the GROMACS package [40]. Both the protein and the polyelectrolyte molecules were modeled through the AMBER all-atom force field [41,42]. Water molecules were described explicitly with the SPCE model [43,44]. The particle mesh Ewald (PME) algorithm with a cutoff distance of 1 nm was used for the evaluation of the electrostatic interactions, whereas non-bonded interactions were parameterized through a spherically truncated 6–12 Lennard–Jones potential, with a cutoff distance of 1 nm and standard Lorentz–Berthelot mixing rules. The time step for the integration of the equations of motion was 1.0 fs. Molecular dynamics simulations were performed in the isothermal–isobaric (NPT) statistical ensemble. The pressure was kept constant at *p =* 1 atm, using the Berendsen barostat [45], while the stochastic velocity rescaling thermostat [46] was used for keeping temperature at the imposed value. Periodic boundary conditions were applied in all three directions.

An additional run of quenching from 368 K to 298 K was executed with a cooling rate equal to 0.8 K/ns. This procedure replicated an experimental protocol involving a specific thermal treatment which was as follows: Protein and polyelectrolyte were mixed at room temperature, and subsequently, the temperature was elevated to 353 K and then allowed to cool to its initial temperature. Hydrophobic regions of the protein were exposed to water at elevated temperature, which upon cooling reorganized and resulted in a possibly different binding to polyelectrolyte.

### 2.2. Experimental Details

Lysozyme from chicken egg in powder form and polyacrylic acid (PAA) M_w_~4000 (PDI ~2) 45 wt. % in H_2_O were obtained from Sigma-Aldrich. Stock aqueous solutions of each component were prepared at 1 mg mL^−1^ using distilled water. The pH of the solutions was adjusted to 7 by adding appropriate volumes of NaOH (1 M) for PAA and HCl (1 M) for lysozyme. Desired final concentrations and lysozyme/PAA mass ratios were achieved by mixing distilled water with appropriate volumes of the stock solutions under gentle stirring. Complexation was performed at room temperature (298 K). After performing several test measurements, two lysozyme/PAA mass ratios were selected to form lysozyme/PAA complexes, i.e., one with low PAA content (r_m_ = c_PAA_/c_Lysozyme_ = 0.01) and one with high PAA content (r_m_ = 0.03). These r_m_ values were motivated by previous works on polyelectrolyte/protein interactions [47,48], where light scattering showed that maximum interaction occurs in the regime of stoichiometric bulk charge neutrality. For the system under study, LYS HAS +8 net charge and PAA monomer have −0.5 net charge (assuming charging of every second monomer) and by using their molar masses at about 14,000 and 70 g mol^−1^, charge neutrality is expected to be rm~0.08. The values of r_ms_ at 0.01 and 0.03 were decided to avoid very strong aggregation and possible macroscopic phase separation. The PAA solution was first added to distilled water, followed by the addition of the lysozyme solution. In addition to the complexes, a sample without PAA (r_m_ = 0) was also prepared. This lysozyme sample allowed for comparison and assessment of how PAA affects the protein’s secondary structure and the strength of the interaction between lysozyme and PAA. Both complexes and the reference sample were measured without and after thermal treatment at pH 7. The thermal treatment involved placing Eppendorf tubes containing the samples in an oven at 353 K for 0.5 h and then leaving them at 298 K until equilibration. These samples are referred to as thermally treated. All experiments were performed at 298 K.

#### 2.2.1. Fourier Transform Infrared Spectroscopy

Fourier transform infrared spectroscopy (FTIR) experiments were conducted using a Bruker Equinox 55 Fourier Transform Instrument (Ettlingen, Germany) equipped with an attenuated total reflectance (ATR) diamond accessory (SENS-IR Technologies, Danbury, CT, USA). A drop of the solution was placed at the center of the sample holder and dried under nitrogen gas. The measurements were performed over a wavenumber range of 500–5000 cm^−1^ with a resolution of 2 cm^−1^, and 64 scans were recorded.

#### 2.2.2. Circular Dichroism

Circular dichroism (CD) measurements were performed with a Jasco J-815 CD spectrophotometer (JASCO Corporation, Tokyo, Japan) with a Peltier model PTC-423S/15 thermo stabilizing system. The samples were loaded on 1 mm quartz Suprasil cells. Aqueous solutions were appropriately diluted so that lysozyme concentration was at 0.1 mg mL^−1^, which was determined as the optimum value for sufficient CD signal. For each sample, CD spectra were collected as averages of four successive measurements. The secondary structure of the proteins was analyzed and estimated using the BeStSel software v1.3.230210 [49].

## 3. Results and Discussion

### 3.1. Effect of Temperature on the Conformational Properties of Lysozyme

The temperature dependence of lysozyme’s structural properties in the mixed system was investigated through a series of MD simulations at defined temperatures: [298 K; 308 K; 328 K; 348 K; 358 K; and 368 K]. The thermostability of the protein molecule is quantified through two measures, the root mean square deviation (rmsd) and the root mean square fluctuation (rmsf), which are presented in Figure 1. The calculations of rmsd and rmsf were based on the alpha carbon (Cα) atoms of all amino acids in the protein. Values of rmsd in the range of [0.15–0.25] nm suggest stable structure [50,51] (i.e., high degree of similarity with the crystallographic structure), which is the case of all Ts. The low values of rmsd for all tested temperatures indicate thermostability for lysozyme (Figure 1a), whereas temperature increase induces a variety of small, localized conformational fluctuations in a non-systematic way. The rmsf (Figure 1b) is a metric of how much a particular residue fluctuates during the simulation, revealing the ones which correspond to the local changes. Its plot versus the residue number identifies the most mobile residues. Temperature increase triggers small, random conformational changes in certain amino acids, without significantly altering the protein’s overall structure. Characteristic configurations of lysozyme at all temperatures after 200 ns of MD simulation are presented in Appendix A, together with the initial crystallographic structure.

Following a thermal quenching from the highest temperature (368 K) back to room temperature (298 K), the reversibility of any structural changes initially is visualized in Figure 2. A configuration at 368 K (Figure 2a), which constitutes the starting point for the quenching process, the final configuration after the quenching run (298 K) (Figure 2c), and a configuration from a run at 298 K (Figure 2b) are presented. Different colors correspond to different types of secondary structure. For the specific cooling rate, a higher similarity between (a) and (c) snapshots is visually observed, which will be quantified in the following, by a secondary structure analysis. For this analysis, the DSSP algorithm [52] is used, where the characteristic hydrogen bond pattern for each secondary element is utilized as a criterion [53] to identify the hydrogen bonds between the backbone atoms of a protein. DSSP analyzes the secondary structure of the protein and classifies the protein into eight types of secondary structure: coil; β-sheet; β-bridge; bend; turn; α-helix; π-helix; and 3_10_-helix.

The specific structure of lysozyme, i.e., the number of residues in every specific conformation as a function of time, is presented in Figure 3 for 298 K and 368 K. The DSSP analysis for the individual residues as a function of time is shown in Appendix A. The increase or decrease in the appearance of each type of structure at different temperatures is presented in Table 2. Small conformational changes in the secondary structure of lysozyme are reported indicating a thermostable protein. Secondary structure quantification was performed through averaging over the last 100 ns of the trajectory (“equilibrated structure”).

The bottom section of Table 2 summarizes the difference in average values between two temperature extremes: (i) the difference between the lowest (i.e., T = 298 K) and the highest (i.e., T = 368 K) temperatures; (ii) the difference between the starting temperature (368 K) and the final temperature (298 K) after a quenching process (i.e., initial vs. final configuration of the run); and (iii) the difference at 298 K comparing the quenched state with the unquenched state. Although changes are very small and almost within the statistical uncertainties for most of the structures, a rough observation can be reported concerning an almost irreversible change in some structures with temperature increase from 298 K to 368 K and then cooling back to 298 K (i.e., last line of Table 2: coil, 3_10_-helix, π-helix). In Figure 4, the superimposed structures of lysozyme are presented at the three states (368 K; 298 K; and 298 K after quenching), where the small local conformational changes are visualized. However, the overall thermostable character of lysozyme is also illustrated.

### 3.2. Effect of PAA on the Conformational Properties of Lysozyme

In order to investigate the effect of the polymer chains on the conformations of the protein molecule, a series of simulations of pure lysozyme in water were performed and used as reference systems for comparison with the mixtures.

A comparison of rmsd and rmsf measures between the reference and the mixed system indicates a negligible effect of the PAA on the structure of lysozyme (Figure 5). The, rmsds as a function of time are shown in Figure 5a,b for T = 298 K and T = 368 K correspondingly, and the values range between [0.15 and 0.25] nm, which suggest a stable structure at both Ts and for both pure lysozyme as well as in the presence of PAA chains. The rmsfs in Figure 5c,d show that at the high temperature, some flexible residues (i.e., in the range [1–25] and [100–125]), which are detected in the pure system, are suppressed. These local changes at the initial and the final residues of lysozyme are probably induced by the way that PAA chains are attached to it (i.e., approaching sites), which will be further discussed below.

The small conformational changes, observed in the secondary structure of lysozyme in the presence of PAA, are quantified in Table 3 at 298 K and 368 K. The effect of the polymer on the protein structure differs between low and high temperature in the way that some conformations are enhanced or diminished. It is also interesting to observe an opposite effect of PAA on lysozyme’s structure at the low and the high temperature (i.e., π-helix, α-helix, β-sheet, and β-bridge). These changes can be attributed to the way that the two components (protein, polymer) associate, due to energetic interactions and/or hydrogen bonding between specific sites of lysozyme with PAA, further analyzed in Section 3.4. In Appendix A, the various protein structures as a function of time for the reference and the mixed system, respectively, are juxtaposed at 298 K.

### 3.3. Conformational Analysis of the Mixed System of Lysozyme and PAA

The association of lysozyme and PAA is presented in the characteristic snapshot of Figure 6. The protein molecule is surrounded by polymer chains which are attached on specific sites. All measures in the following are calculated in the “steady state” part of the trajectory (i.e., the last 100 ns of the simulation run).

Pair radial distribution functions (rdf), g(r), based on the center of mass of the residues of the protein, of the monomers of the polymer, and of the water molecules, quantify the proximity between the different species in the aqueous solution. The affinity of polymer chains for protein is reduced compared to their self rdfs (i.e., PP, LL), as it is presented in Figure 7a. Moreover, both molecules are hydrophobic, with lysozyme attaining a higher degree of hydrophobicity, as it is observed in the inset of Figure 7a, where rdfs for polymer–water and for protein–water pairs are presented. In Figure 7b, the rdfs between lysozyme and PAA at different temperatures do not show any systematic trend with temperature. In addition, the curves at 298 K with and without quenching are rather close, indicating a similar tendency for association at the two states.

A key question is how lysozyme’s attachment sites for PAA chains are affected by the thermal treatment. For this reason, distances between PAA and lysozyme atoms are calculated and the pair with the minimum distance between protein atoms and all PAA chains’ atoms is established. The specific protein atom is assigned to the corresponding amino acid for each configuration. A further classification is then applied according to a criterion for distances less than 0.35 nm, which corresponds to that which meets the hydrogen bond criterion. Finally, a histogram of the percentage of time of association of lysozyme with PAA, calculated for the amino acids with the shortest approach in polymer chains, is created, ranking the 20 different amino acids that make up lysozyme (association rate). Figure 8 shows histograms at 298 K, 368 K, and 298 K after quenching, where normalization with the number of corresponding amino acid types within the protein molecule and with the number of the configurations of the analyzed trajectory was performed. The inset histograms show the same results where no normalization was applied.

A class of amino acids with the closest approximation to PAA most frequently (i.e., association rate > ~250, based on the inset of Figure 8), is chosen to be analyzed. By observing Figure 8, several insightful observations can be made, which are summarized in Table 4: (a) A dramatic decrease in the association rate with PAA for TYR with a concomitant increase for SER is evident from 298 K to 368 K. Quenching back to 298 K does not recover the former state. (b) ARG maintains an almost constant and remarkably high compared to other amino acids association rate with the polymer chains, which is slightly affected by thermal treatment. (c) For ASN, the quenching seems to reduce the association rate with PAA, although it is not affected by the increase in temperature. (d) Finally, for LYS, a small increase in the association rate with PAA is observed at 368 K, which is reversible, ending almost at the initial value after quenching.

### 3.4. Energetics–Hydrogen Bonds

The role of hydrogen bonds is critical for the complexation between protein and polymer chains. Detection of hydrogen bonding was based on the following geometric criteria associated with an acceptor A and a donor D bonded to a hydrogen atom: (1) the distance r between D and A, r(D...A) ≤ 3.5 Å, and (2) the angle HDA ≤ 30°. Thus, in the next Table 5, hydrogen bonds between PAA and the amino acids with the dominant contribution to the histogram of Figure 8 (i.e., association rate > ~250) are counted and compared at the three temperature states. Values are divided with the number of the corresponding amino acids within the protein. The number of hydrogen bonds follows the observations reported above in Table 4. The closer the amino acids to PAA, the more the hydrogen bonds, with the ARG keeping the lead, almost unaffected by the temperature treatment. Increased hydrogen bonding between ARG and PAA makes ARG a stable anchor point for lysozyme on PAA, as it also results from their association rate.

An overall picture for the effect of the temperature on the hydrogen bonding in the system is given in Figure 9 and Appendix A, where hydrogen bonds between all pairs of molecules in the system are presented (i.e., PAA-PAA (PP); PAA-water (PW); lysozyme-lysozyme (LL); lysozyme-water (LW); and PAA-lysozyme (PL)). The total number of hydrogen bonds formed between polymer and protein are not affected neither by temperature increase nor by quenching. Although these values fluctuate within error bars, conformational rearrangements impose local increase/decrease in hydrogen bonding of specific amino acids with PAA, indicating local structural changes of lysozyme, as discussed above (Figure 8 and Table 4). Comparing the number of hydrogen bonds that water forms with lysozyme and with PAA, higher hydrophobicity of the protein is indicated.

In addition, temperature increase from 298 K to 368 K induces an obvious reduction in the hydrogen bond network of water with PAA. This is attributed to the reduced solvent accessible surface area (sasa) of polymer chains, presented in Table 6, as a result of their temperature-induced conformational changes. The systematic reduction in sasa with temperature increase is irreversible after quenching. A much smaller reduction in hydrogen bonding is observed between lysozyme and water; however, in this case, this is not related to any change in sasa (Table 6). This can be attributed to a rise in the concentration of Na^+^ cations near the protein surface with increasing temperature. Cations occupy water sites with hydrogen bonding potential, causing this decrease in the number of water—lysozyme hydrogen bonds. This is evident in the pair radial distribution function calculated between the Na^+^ cations and the protein atoms and presented in Appendix A at 298 K, 368 K and after quenching back to 298 K. The peak intensity in the rdf curve is greater at 368 K compared to 298 K, while it weakens again after quenching. For all other pairs, hydrogen bonding is unaffected by temperature and independent from the temperature treatment.

Energetic components, in a synergistic manner with hydrogen bonds, determine the conformational characteristics of the protein–polymer complex. For this, both electrostatic and van der Waals interactions between PAA-lysozyme atoms, PAA-PAA atoms and lysozyme-lysozyme atoms were calculated and average values over the “steady state” part of the trajectory are presented in Figure 10 and Figure 11, respectively, as a function of temperature.

No systematic trend of protein–protein Coulombic interactions is observed with temperature increase from 298 K to 368 K, although very similar energies are found with and without quenching at 298 K (Figure 10a). On the other hand, van der Waals interactions weaken at higher temperature values (i.e., beyond 328 K), with no further change up to 368 K, while quenching at 298 K reverses this behavior (Figure 11a). For protein–polymer interactions, both Coulomb and Lennard Jones energies fluctuate with no specific temperature effect. Coulombic energy values are very close at 298 K with and without quenching (Figure 10c), while quenching leads to more attractive van der Waals interactions compared to the corresponding energy values at 298 K (Figure 11c). Finally, a systematic response to the temperature gradient is observed only on PAA, with a weakening of electrostatic attraction (Figure 10b) and strengthening of van der Waals attraction (Figure 11b) with increasing temperature. Values for both energetic contributions are close at 368 K and after quenching to 298 K, indicating that conformational changes induced on PAA chains with temperature increase are irreversible. Furthermore, Coulombic interactions dominate over the van der Waals ones between all species.

Overall, thermal treatment does not affect the complex between PAA and lysozyme to a large extent, in terms of energetics or hydrogen bonding. However, it induces local conformational changes related to the approaching sites between the two species, which are mainly attributed to the irreversible effect of temperature on the PAA. In addition, although lysozyme is thermostable as a whole, specific amino acids are affected by the increase in temperature and the presence of the polyelectrolyte in their vicinity, inducing small rearrangements. As a result, a different complexation pathway is followed.

## 4. Experimental Evidence

FTIR is sensitive to specific bond vibrations associated with secondary structures and CD is sensitive to the chiral nature of protein structures. These techniques are typically used to determine protein conformational transitions [5,55]. However, the spectroscopic characterization of protein secondary structure is often partially unreliable when samples are not extremely pure and abundant [56]. Therefore, these two methods are combined to provide a comprehensive view of the protein secondary structure under different solution conditions and upon interaction with other components [57]. FTIR experiments were conducted to identify the conformation of lysozyme in the free state and in the complexes before and after thermal treatment.

According to Chang’s study [58], native lysozyme remains completely stable at 328 K for up to 60 min. However, at a higher temperature of 338 K, approximately 55% of the native of lysozyme partially unfolds under similar conditions. The initial rate of lysozyme partial unfolding increases nearly 14-fold as the temperature rises from 338 K to 348 K. The analysis further indicates moderate structural changes of lysozyme within 20 min at a temperature of 348 K. In addition, Xu et al. [21] reported that mild denaturation, self-aggregation, and phase separation occurred when the temperature exceeded 343 K. In this context, a thermal treatment temperature of 353 K was selected, which is slightly above the temperature that lysozyme partially unfolds. The amide I band (1700–1600 cm^−1^) (Figure 12a) was investigated as it contains information on the various structural components [13]. The FTIR data were modeled using a superposition of Gaussian functions [59,60]. The deconvolution of the amide I signal required the use of seven terms (Figure 12b). The optimal positions of the Gaussian peaks were assigned to the various structural components as detailed in Appendix A. Appendix A presents the results of the estimation of the secondary structure of native lysozyme at pH around 7 from various previous studies [61,62,63,64,65]. The contribution of each conformation appears to vary slightly across different studies as there are small differences in the selected assignments. Our results show very good agreement with the recent study of Sadat et al. [62], where the same assignments with our study were used.

Therefore, the values 47% for α-helix, 18% for β-sheet, and 35% for β-turn (Figure 13) for lysozyme in the free state without thermal treatment are considered reliable as a starting point for investigation in the several conditions of this study.

In lysozyme, there is an increase in absorption intensity at wavenumbers ~1600–1630 cm^−1^ after thermal treatment indicating the formation of additional β-sheet structures (Figure 12a). Indeed, there is an increase in β-sheet to 24%, which is accompanied by a decrease in α-helix to 22% (Figure 13). β-turn is found to increase to 54%. These findings suggest the irreversible partial unfolding of lysozyme globular conformation to a more open structure with the possible formation of intermolecular β-sheets. The complexation of lysozyme with PAA results in a decrease in α-helix to 25 and 26% and an increase in β-sheet to 27 and 21% for r_m_ 0.01 and 0.03, respectively. Regarding the FTIR signal, the addition of PAA shifts the peak to slightly lower wavenumbers, indicating an increase in β-sheet and a decrease in α-helix content. There is a decrease in β-turn to 30 and 33%; however, most of the α-helix loss leads to the appearance of random coil with contribution 18 and 20% for r_m_ 0.01 and 0.03, respectively. It is evident that the interaction of lysozyme with PAA leads to a partial disruption of the helices towards a random conformation. Upon thermal treatment, there is a marked increase in the FTIR absorbance in the region corresponding to β-sheet structures. A-Helix decreases to 22 and 18%, β-sheet increases to 33 and 40%, and β-turn decreases to 27 and 25% for r_m_ 0.01 and 0.03, respectively. Random coil remains the same for r_m_ 0.01 and decreases to 17% for r_m_ 0.03. This shows that β-sheet formation is induced by thermal treatment also within the complexes. This effect was used in the past from members of our group for the stabilization of protein/polysaccharide NPs [47,66,67].

According to the CD experiments, in the secondary structure of lysozyme in the native state α-helix (29%) is much higher than β-sheet (9%), which is in qualitative agreement with FTIR analysis (Figure 13). Random conformation (which includes β-turns) has a percentage of 62%. Representative fitting curves in comparison with experimental data can be found in Appendix A. The secondary structure as it is observed in CD undergoes a slight change after thermal treatment (Figure 14). The addition of a small amount of PAA (r_m_ = 0.01) does not seem to significantly affect the protein structure (Figure 14a). However, after thermal treatment a minor difference in the protein structure is observed between r_m_ = 0 and r_m_ = 0.01. When the amount of PAA increased further (r_m_ = 0.03), a remarkable change in the spectrum and consequently in the protein conformation is evident, which is more pronounced in the thermally treated sample (Figure 14b). Thus, for free lysozyme, thermal treatment results in a decrease in the α-helix structure from 29 to 27 and an accompanying increase in the β-sheet structure from 9 to 14. The addition of PAA results in a decrease in a-helix to 13% for the high r_m_ (although this decrease is not observed for the low r_m_) and a simultaneous increase in β-sheet to 11 and 27% for r_m_ 0.01 and 0.03, respectively. In thermally treated complexes α-helix dropped to 17% for r_m_ 0.01 and was diminished for r_m_ 0.03 while β-sheet was at 25 and 45% for r_m_ 0.01 and 0.03, respectively. It is evident that the results from CD support the ones of FTIR as they show clearly the main effects of partial protein unfolding because of thermal treatment and complexation, which are the decrease in α-helix and the increase in β-sheet conformation.

## 5. Linking MD Insights to Experimental Data and Broader Scientific Impact

Molecular simulations provide a finer classification of protein structures compared to experimental techniques. Therefore, eight different types of structures emerge from the secondary structure analysis (Figure 3 and Table 2), with some of them seldom detectable (i.e., π-helix, 3_10_-helix) in experiments. On the contrary, FTIR [68] and CD [69] spectroscopies mainly assign the α-helical structures, while other helical structures are characterized as “random coil” or “others”. Therefore, the comparison between the two approaches is mostly qualitative rather than quantitative.

There appears to be a rough agreement in the initial percentages of the various structures, making an assumption to add up the different types of helices, found with DSSP analysis, in one (38.8%), and comparing with the a-helix detected experimentally, and the same for β-sheet added to β-bridge (10.4%), whereas for turn the percentage is (25.5%). Note here that the classification is based on the 129 residues of the model protein, which corresponds to the 100% of the protein structure. Coil and bend are two additional structures detected in the model system but not in the experiment, which can be responsible for any remaining differences. According to the above assumption an increasing order of model detected structures is as follows: “sum of all the remaining structures” > α-helix > β-sheet, in line with both FTIR and CD measurements. Concerning the effect of thermal treatment on the complex, simulations show a small decrease in the helices to 37.6%, but no change is detected to “β-sheet and β-bridge” conformations, at the specific cooling rate. However, changes exist on each individual structure as depicted in Table 2, resulting in local rearrangements of protein, which explain in a detailed manner the partial change in structure observed experimentally.

Furthermore, the MD calculations can be used for the development and optimization of new materials. The knowledge of the number and the kind of amino acids that take part in attachments with the polyelectrolyte give a very clear picture of the ability of the protein globules to connect with the polyelectrolyte to create nanoclusters and 3D nano or macroscopic networks. It also allows for knowing and designing which amino acids are left for interactions with other compounds, e.g., other proteins, drugs, and nutrients. The information on the conformational changes in the protein from its native state can be used to optimize the desirable biomaterial, according to the needs of the application, in terms of functionality versus number of bonds and network structure. Similarly, the intensity of the thermal treatment could be potentially tuned to balance between the enhancement of structural stability of the nanostructures and the compromise of the functionality of the protein which is related to its native state.

## 6. Conclusions

We use all-atom MD simulations combined with experimental techniques to investigate the complexation between PAA and lysozyme at the atomic level. The focus is on the effect of PAA and temperature on the conformational properties of lysozyme. In addition, a specific thermal process (relevant to stabilization of the complexes with non-electrostatic bonds) is explored for the way that it influences the interactions and the complexation of these molecules in water. Our analysis reveals subtle changes in the secondary structure of lysozyme, suggesting a high degree of thermal stability. While most structural variations fall within the range of statistical uncertainty, some structural elements appear to undergo a partially irreversible change after the thermal treatment, in qualitative agreement with experimental findings where a temperature-induced, partially irreversible transition of lysozyme’s globular conformation towards a more open structure is observed. Atomic details relating conformational changes to energetic contributions and hydrogen bonding highlight the origin of partial protein unfolding.

By analyzing the protein’s structure and the proximity of the molecules within the complex, we can determine their specific arrangement and how they interact. Specific recognition sites where the polyelectrolyte binds to the protein are revealed based on a detailed analysis for the amino acids with the shortest approach to polymer chains at 298 K, 368 K, and 298 K after quenching. The major findings are the following: (a) Compared to other amino acids, arginine (ARG) shows a remarkably high and temperature-independent association rate with polymer chains. (b) At higher temperatures (368 K), tyrosine (TYR) shows a significant decrease in its tendency to approach the polymer compared to 298 K, while serine (SER) shows the opposite trend. This change is irreversible upon cooling back to 298 K; (c) lysine (LYS) shows a small increase in its approach towards the polymer at higher temperatures. This increase is reversible, returning nearly to its initial value after quenching. (d) Quenching the system reduces the frequency of asparagine (ASN) approaching the polymer chains.

Energy and hydrogen bond calculations highlight the driving forces of the aforementioned behavior. Interestingly, the number of hydrogen bonds between the protein and polymer stays the same regardless of temperature or thermal process. Despite the minimal impact on global conformation, temperature-induced rearrangements lead to localized variations in hydrogen bonding between specific amino acids and polymer. This aligns with the observations regarding the proximity between lysozyme’s amino acids and PAA. The enhanced hydrogen bonding which is observed between ARG and PAA solidifies ARG’s role as a stable binding point for lysozyme on PAA.

The affinity between the two molecules is quantified through the calculation of both electrostatic and van der Waals interactions. The major role is played by electrostatic interactions, which are stronger than the weak van der Waals between all pairs. The strength of attractions between the protein and polymer (both Coulombic and Lennard Jones) fluctuates with temperature, but there is no clear trend overall. PAA is most affected by temperature where electrostatic attraction between PAA chains weakens with temperature increase, whereas van der Waals attraction strengthens and this change is irreversible upon cooling.

Experimental results from circular dichroism (CD) for lysozyme and its complexes with PAA align well with the Fourier transform infrared spectroscopy (FTIR) data. Both techniques reveal a decrease in alpha-helix content and an increase in beta-sheet structure, which suggests that the thermal treatment and complexation process leads to partial unfolding of the protein.

Beyond explaining the protein’s conformational changes under these specific conditions, the atomistic detail, revealed by the MD simulations, provides invaluable guidance for future experiments. This computational analysis offers a framework for interpreting existing or new spectroscopic data on protein conformational changes observed in other polyelectrolyte/protein systems and under analogous thermal treatments.

## Figures and Tables

**Figure 1 polymers-16-02565-f001:**
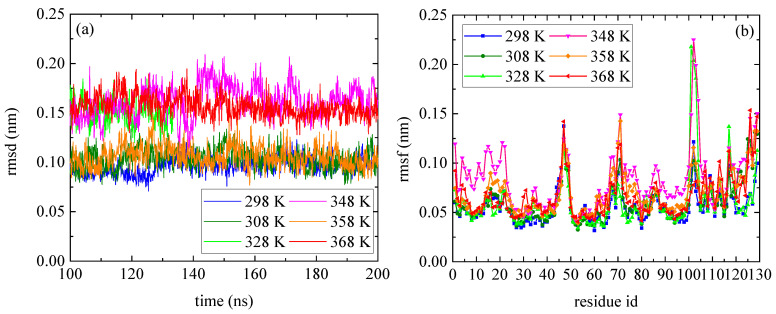
(**a**) Root mean square deviation (rmsd) and (**b**) root mean square fluctuation (rmsf) for lysozyme at different Ts (298–368) K from the last 100 ns of the trajectory.

**Figure 2 polymers-16-02565-f002:**
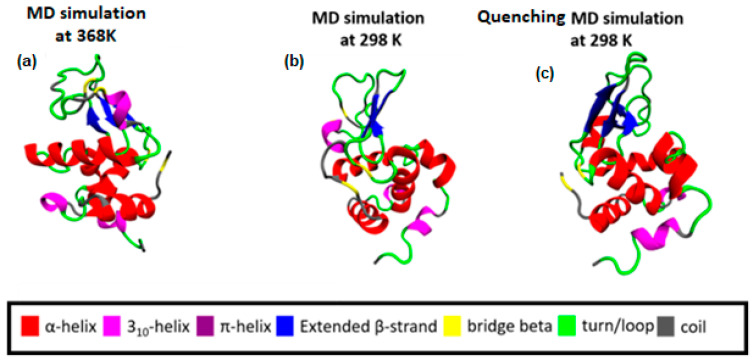
(**a**) Initial configuration of Lysozyme for the quenching MD run, which is the output from the MD simulation at 368 K; (**b**) The final configuration of Lysozyme from MD simulation at 298 K. (**c**) The final configuration of Lysozyme from the quenching MD run (298 K).

**Figure 3 polymers-16-02565-f003:**
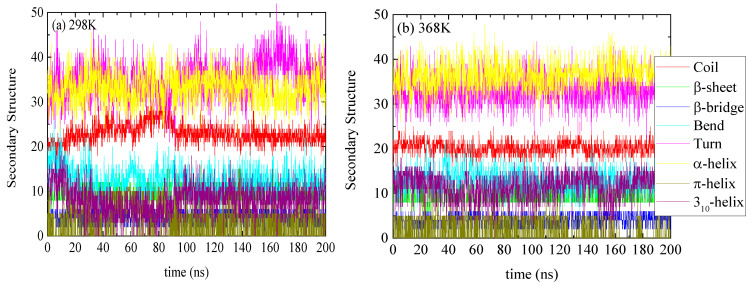
Secondary structure quantification as a function of time for lysozyme at (**a**) 298 K and (**b**) 368 K; y-axis represents the number of residues in each structure which sum up to the total number of residues of lysozyme (129).

**Figure 4 polymers-16-02565-f004:**
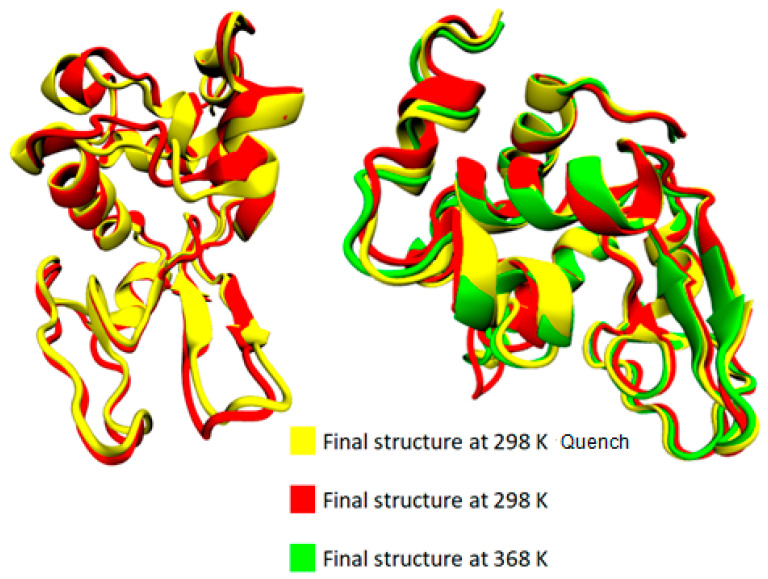
Superimposed structures from VMD [54] at 368 K, 298 K, and 298 K quench.

**Figure 5 polymers-16-02565-f005:**
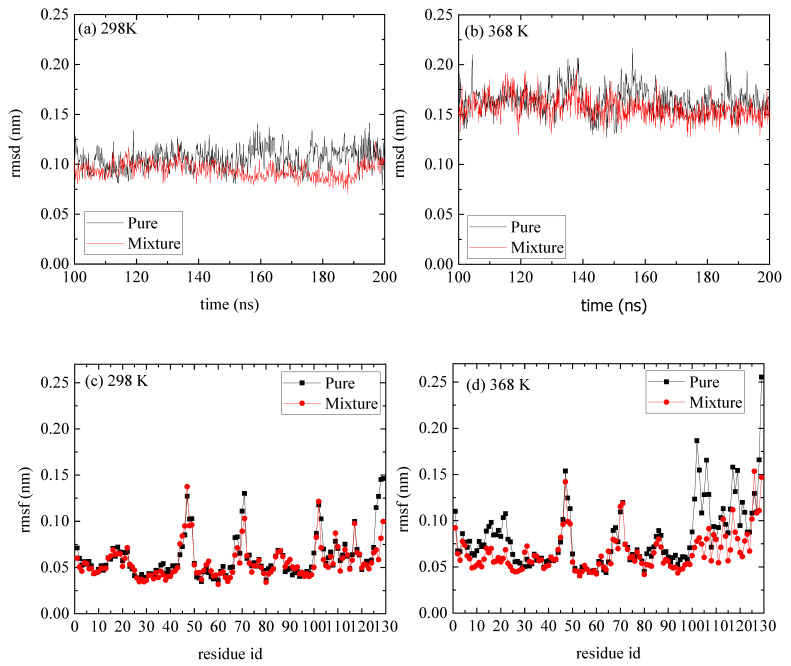
(**a**,**b**) Root mean square deviation (rmsd) of lyosozyme at 298 K and 368 K, respectively, for systems of lysozyme in water and of PAA/lysozyme mixtures; (**c**,**d**) root mean square fluctuation (rmsf) at 298 K and 368 K, respectively, from pure lysozyme in water and from mixture from the last 100 ns of the trajectory.

**Figure 6 polymers-16-02565-f006:**
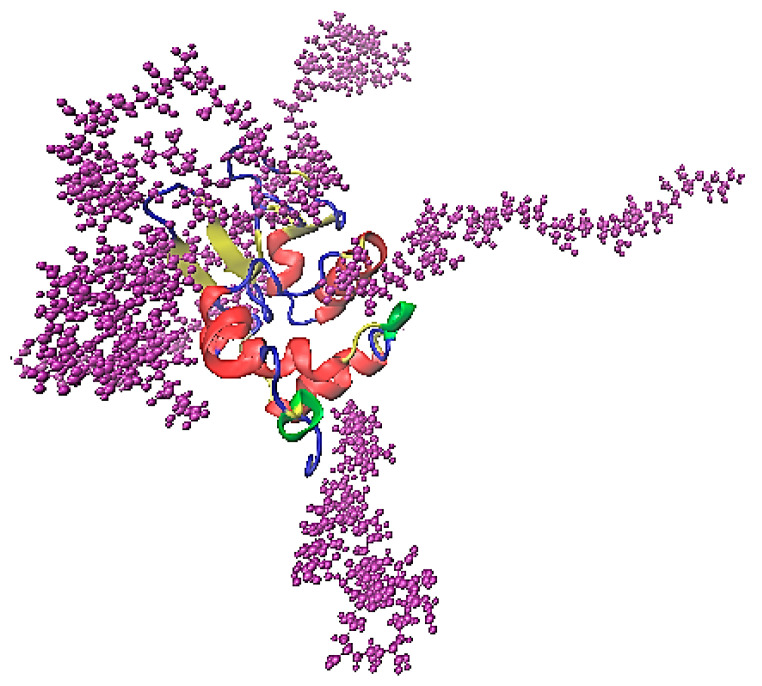
A characteristic snapshot from the simulation model for a complex of lysozyme with PAA in aqueous solution. Water molecules are omitted for clarity.

**Figure 7 polymers-16-02565-f007:**
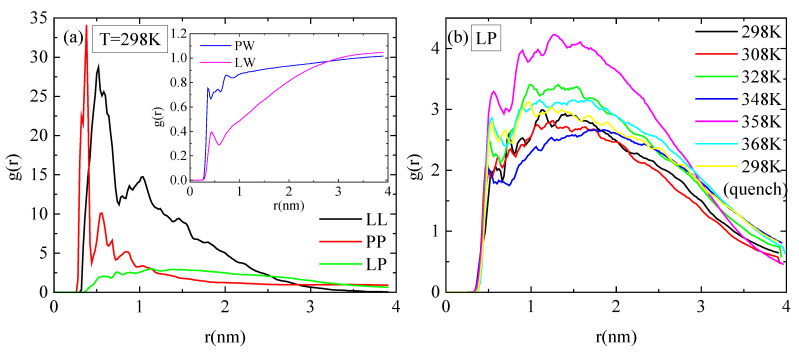
Pair radial distribution functions, g(r), based on the center of mass of residues for the protein and the center of mass of monomers for the polymer chains or water molecules. (**a**) All possible pairs at 298 K; (**b**) temperature dependence for the lysozyme/PAA component.

**Figure 8 polymers-16-02565-f008:**
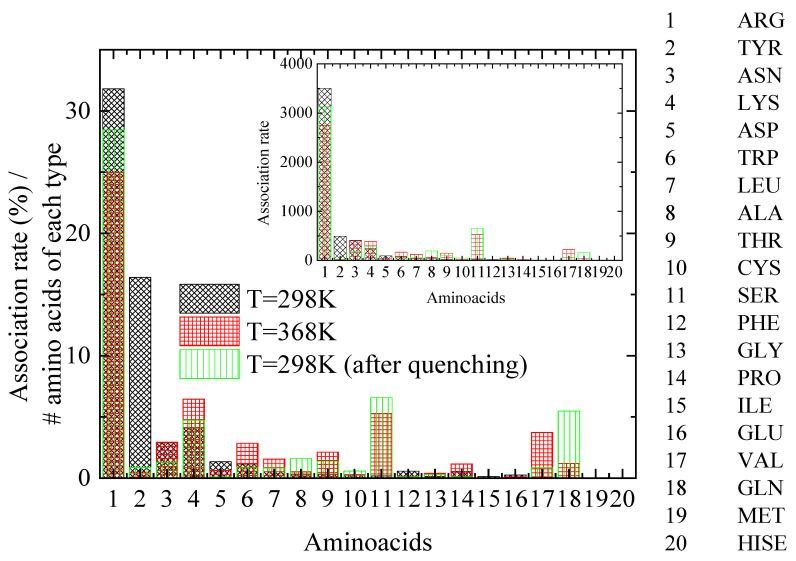
Histogram of the association rate of lysozyme with PAA, calculated for the amino acids with the closest distance to polyelectrolyte (PAA) chains at the “steady state” part of the trajectory. Inset: Histograms of same results where no normalization was applied.

**Figure 9 polymers-16-02565-f009:**
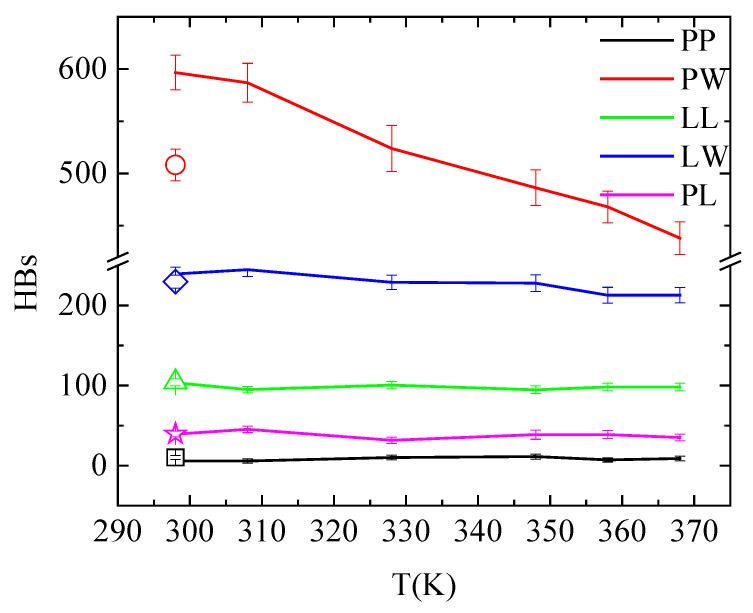
Average number of hydrogen bonds over the “steady state” part of the trajectory between all possible pairs of molecules in the system as a function of temperature. Open symbols stand for the corresponding values after quenching.

**Figure 10 polymers-16-02565-f010:**
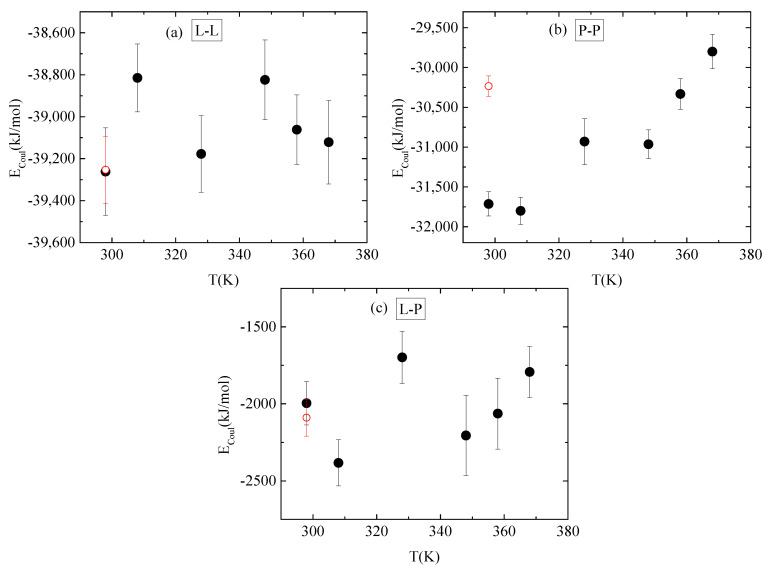
Average electrostatic interactions between (**a**) lysozyme-lysozyme; (**b**) PAA-PAA; and (**c**) lysozyme-PAA as a function of temperature. Red open symbol corresponds to 298 K after quenching.

**Figure 11 polymers-16-02565-f011:**
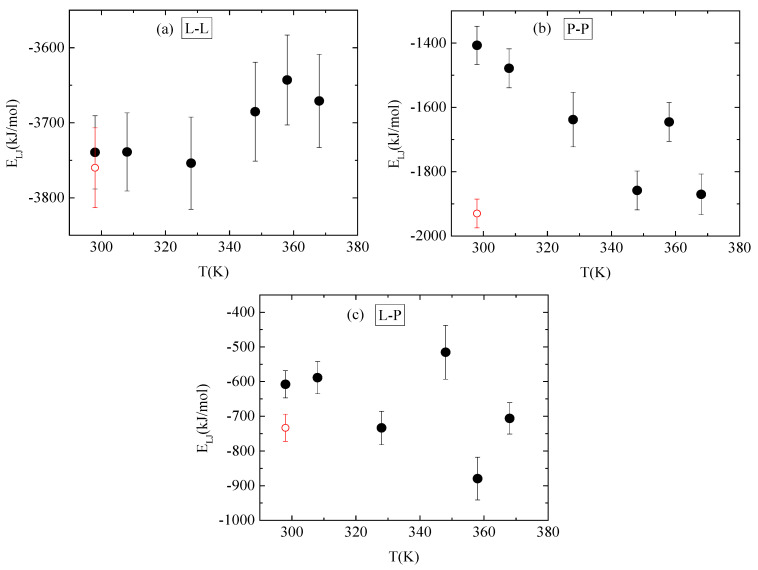
Average van der Waals interactions between (**a**) lysozyme-lysozyme; (**b**) PAA-PAA; and (**c**) lysozyme-PAA as a function of temperature. Red open symbol corresponds to 298 K after quenching.

**Figure 12 polymers-16-02565-f012:**
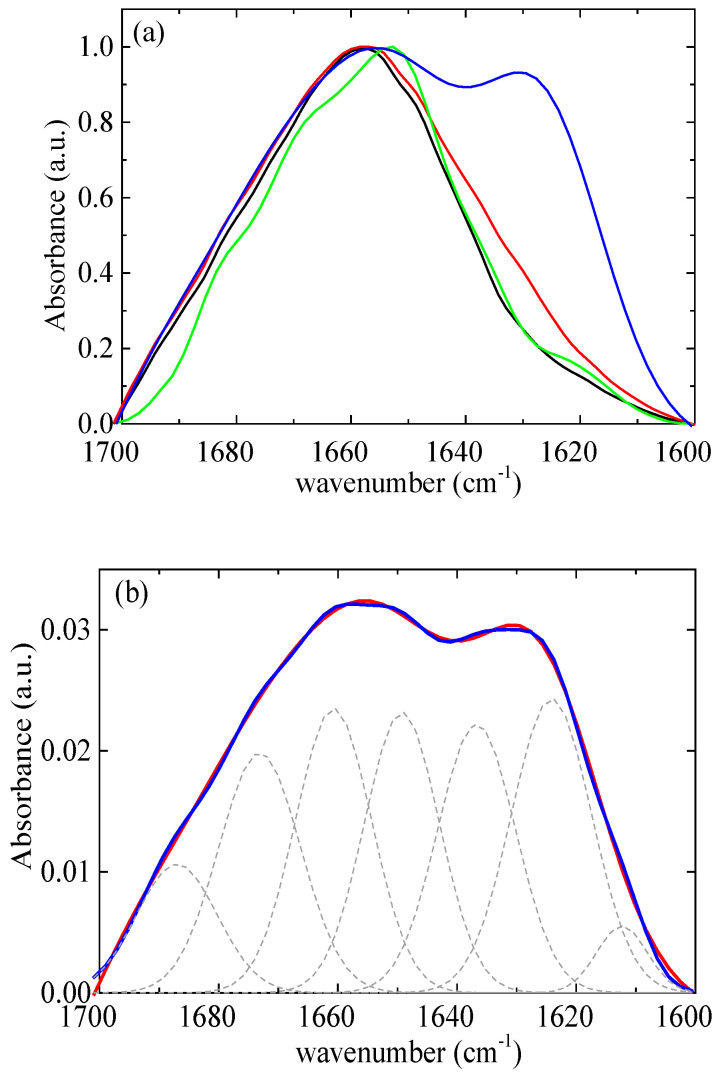
(**a**) FTIR absorbance spectra from lysozyme at pH 7 (black), at pH 7 thermally treated (red), and from r_m_ = 0.03 at pH 7 (green), at pH 7 thermally treated (blue). (**b**) FTIR absorbance spectra (red) from complexes at r_m_ = 0.03 after thermal treatment. Best fits are shown in blue and separate contributions in dashed black.

**Figure 13 polymers-16-02565-f013:**
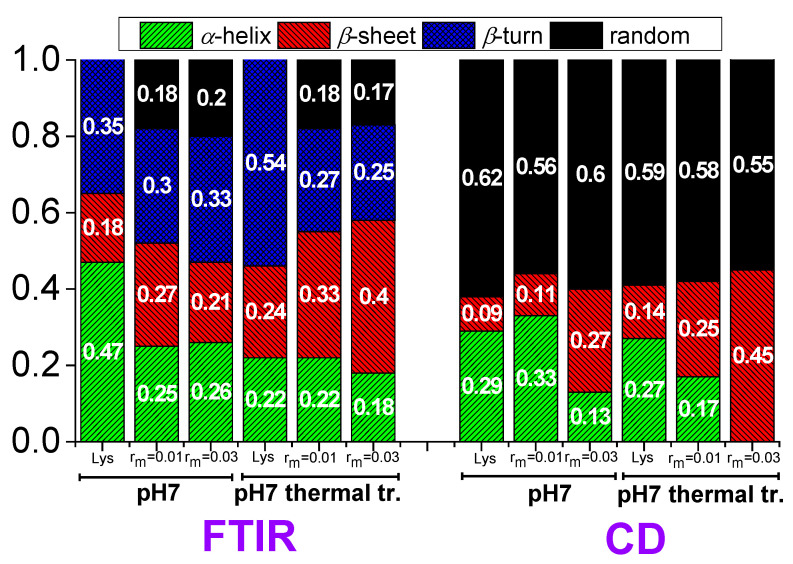
Contributions in the secondary structure of lysozyme extracted from FTIR and CD in free state and in the complexes.

**Figure 14 polymers-16-02565-f014:**
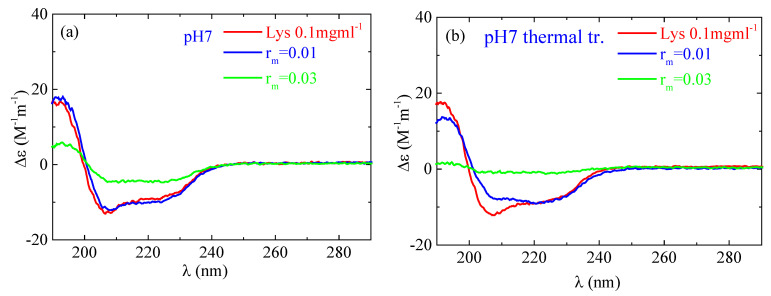
CD spectra from the lysozyme (red) and complexes with r_m_ = 0.01 (blue) and r_m_ = 0.03 (green) (**a**) without and (**b**) with thermal treatment at pH 7.

**Table 1 polymers-16-02565-t001:** Details of the simulated systems.

Systems	PAA	Lysozyme	Total Atoms	Water Molecules	Ions (Na^+^)
**Reference**	-	1	33,798	10,610	(Cl^−^) 8
**Mixture**	5	1	50,490	15,576	(Na^+^) 92

**Table 2 polymers-16-02565-t002:** Secondary structure of lysozyme at various temperatures (quantification over the last 100 ns). Values correspond to the number of residues in each structure which sum up to the total number of residues of lysozyme (129).

Systems	Coil	β-Sheet	β-Bridge	Bend	Turn	α-Helix	π-Helix	3_10_-Helix
**L_298_**	20.18 ± 1.46	9.25 ± 1.87	4.15 ± 1.41	12.46 ± 2.39	32.92 ± 3.22	36.75 ± 2.85	1.32 ± 2.35	11.96 ± 2.90
**L_308_**	20.01 ± 1.50	9.06 ± 1.80	3.85 ± 1.18	13.98 ± 2.69	30.28 ± 2.89	36.44 ± 2.68	1.45 ± 2.42	13.93 ± 2.33
**L_328_**	22.91 ± 1.72	8.11 ± 0.84	4.56 ± 1.19	13.35 ± 2.27	31.97 ± 2.96	37.79 ± 3.43	4.09 ± 3.28	6.22 ± 2.45
**L_348_**	20.68 ± 1.80	9.02 ± 1.78	4.02 ± 1.36	14.43 ± 2.87	33.25 ± 3.92	34.53 ± 3.49	2.02 ± 2.54	11.04 ± 3.28
**L_358_**	19.73 ± 1.55	8.29 ± 1.08	4.39 ± 1.11	14.79 ± 2.76	32.35 ± 3.73	35.31 ± 2.60	1.26 ± 2.31	12.88 ± 2.92
**L_368_**	22.38 ± 1.64	9.04 ± 1.82	4.30 ± 1.41	11.85 ± 2.61	36.35 ± 4.31	33.11 ± 3.39	3.33 ± 3.42	8.64 ± 2.86
**L_298Quench_**	22.03 ± 1.46	9.35 ± 1.91	4.17 ± 1.39	12.07 ± 2.29	32.77 ± 2.99	39.07 ± 2.68	0.53 ± 1.59	9.00 ± 2.99
**Variation in conformations (V2 − V1)/V1**								
**V1: 298 vs. V2: 368**	0.11	−0.02	0.04	−0.05	0.10	−0.10	1.52	−0.28
**V1: 368 vs. V2: 298Quench**	−0.02	0.03	−0.03	0.02	−0.09	0.18	−0.84	0.04
**V1: 298 vs. V2: 298Quench**	0.09	0.01	0.005	−0.03	−0.004	0.06	−0.59	−0.25

**Table 3 polymers-16-02565-t003:** Secondary structure quantification of lysozyme at 298 K and 368 K.

Variation in Conformations (Mixed-Pure)/Pure								
Systems	Coil	β-Sheet	β-Bridge	Bend	Turn	α-Helix	π-Helix	3_10_-Helix
**Pure vs. Mixed @298 K**	0.016	0.025	−0.177	−0.151	0.066	0.035	−0.444	0.028
**Pure vs. Mixed @368 K**	0.012	−0.024	0.109	−0.186	0.061	−0.014	0.695	−0.078

**Table 4 polymers-16-02565-t004:** Amino acids with high association rate with PAA. (Bold values highlight a significant effect of temperature).

	Association Rate (%)					
#Residues within Lysozyme	T = 298 K		T = 368 K		T = 298 K (after Quenching)	
11	ARG	31.81	ARG	24.98	ARG	28.45
3	TYR	**16.4**	TYR	**0.43**	TYR	**0.87**
14	ASN	2.87	ASN	2.94	ASN	1.31
6	LYS	4.1	LYS	6.45	LYS	4.73
10	SER	**0.18**	SER	**5.28**	SER	**6.57**

**Table 5 polymers-16-02565-t005:** Average number of hydrogen bonds per residue over an “steady state” part of the trajectory for the residues listed as closest to the polymer most frequently.

	Hydrogen Bonds Residue—PAA/Residue		
Residue	T = 298 K	T = 368 K	T = 298 K (after Quenching)
ARG	2.51 ± 0.27	1.96 ± 0.28	2.48 ± 0.21
TYR	0.48 ± 0.18	0.10 ± 0.18	0.004 ± 0.04
ASN	0.32 ± 0.12	0.24 ± 0.11	0.14 ± 0.06
LYS	0.39 ± 0.14	0.64 ± 0.19	0.47 ± 0.21
SER	0.06 ± 0.05	0.27 ± 0.12	0.32 ± 0.14

**Table 6 polymers-16-02565-t006:** SASA over the “steady state” part of the trajectory for PAA and lysozyme at different temperatures.

Systems	SASA (nm^2^)—PAA	SASA (nm^2^)—Lysozyme	T (K)
L_298_	162.48 ± 3.28	70.76 ± 1.13	298
L_308_	159.48 ± 3.90	72.74 ± 1.12	308
L_328_	146.70 ± 4.34	70.06 ± 1.16	328
L_348_	131.12 ± 2.44	70.54 ± 1.28	348
L_358_	137.10 ± 3.54	72.23 ± 1.24	358
L_368_	130.31 ± 3.69	70.92 ± 1.33	368
L_quenching_	132.52 ± 1.41	70.34 ± 1.08	298

## Data Availability

Data is contained within the article or Appendix A.

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
