# Peer review of "Exploring the Origins of Association of Poly(acrylic acid) Polyelectrolyte with Lysozyme in Aqueous Environment through Molecular Simulations and Experiments"

_polymers, 2024, doi:10.3390/polym16182565_

Round 1

Reviewer 1 Report

Comments and Suggestions for Authors

Congratulations for this fine work.

I have observations regarding only the introduction and conclusion parts, which are too long.

Even if it presents in detail the current state of research in this field, please find a way to summarize this information in a shorter and more concise form.

Similarly, the conclusions part is too extensive. Many of the statements from this section can be introduced in the experimental part. Thus, the conclusions become clearer and highlight the purpose of this research and the obtained results.

So, please reformulate the introduction and conclusion parts in a shorter form, for a better reading and understanding experience.

Author Response

A file is attached

Reviewer 2 Report

Comments and Suggestions for Authors

The article explore the polymer/protein interactions that is of great importance for biomedicine and it enables development of modern therapeutic drugs and treatment strategies. The article is well structured and written and in my opinion it will be good contribution to this research field. However some issues have to be addressed before publishing:

1.      Why in simulation studies a mass ratio Lys/PAA of 1 was selevcted while at the experimental they are rm 0.01 and 0.03? How the rm 0.01 and 0.03 are calculated and why they are associated with low and high PAA content? This need to be clarified.

2.      It is not clear at what temperature the complexes are prepared at the experimental study.

3.      How the temperature 368K was selected? Probably the authors should mentioned what is the behavior of the investigated objects at 310K, that is the body temperature?

4.      Both C and K are used as temperature units. It is proper to use only one of them.

5.      In the results section p. 4 Experimental Evidence the term "thermally treated" have to be clarified. The Figure 12 caption as well.

6. In the text of p. 4. Experimental Evidence it is not clearly stated when for what rm is discussed. This is confusing for the reader to follow the discussion.

Author Response

A file is attached
